# Melt-Pool Dynamics and Microstructure of Mg Alloy WE43 under Laser Powder Bed Fusion Additive Manufacturing Conditions

Julie Soderlind [1,2], Aiden A. Martin [2], Nicholas P. Calta [2], Philip J. DePond [2], Jenny Wang [2], Bey Vrancken [2], Robin E. Schäublin [3], Indranil Basu [3], Vivek Thampy [4], Anthony Y. Fong [4], Andrew M. Kiss [4], Joel M. Berry [2], Aurélien Perron [2], Johanna Nelson Weker [4], Kevin H. Stone [4], Christopher J. Tassone [4], Michael F. Toney [4,5], Anthony Van Buuren [2], Jörg F. Löffler [3], Subhash H. Risbud [1] and Manyalibo J. Matthews [2,*]

1   Department of Materials Science and Engineering, University of California,
    Davis, CA 95616, USA
2   Physical and Life Sciences Directorate, Lawrence Livermore National Laboratory,
    Livermore, CA 94550, USA
3   Laboratory of Metal Physics and Technology, Department of Materials,
    ETH Zurich, 8093 Zurich, Switzerland
4   Stanford Synchrotron Radiation Lightsource, SLAC National Accelerator Laboratory,
    Menlo Park, CA 94025, USA
5   Department of Chemical and Biological Engineering, and the Renewable and Sustainable
    Energy Institute, University of Colorado, Boulder, CO 80309, USA
*   Correspondence: matthews11@llnl.gov

**Abstract:** Magnesium-based alloy WE43 is a state-of-the-art bioresorbable metallic implant material. There is a need for implants with both complex geometries to match the mechanical properties of bone and refined microstructure for controlled resorption. Additive manufacturing (AM) using laser powder bed fusion (LPBF) presents a viable fabrication method for implant applications, as it offers near-net-shape geometrical control, allows for geometry customization based on an individual patient, and fast cooling rates to achieve a refined microstructure. In this study, the laser–alloy interaction is investigated over a range of LPBF-relevant processing conditions to reveal melt-pool dynamics, pore formation, and the microstructure of laser-melted WE43. In situ X-ray imaging reveals distinct laser-induced vapor depression morphology regimes, with minimal pore formation at laser-scan speeds greater than 500 mm/s. Optical and electron microscopy of cross-sectioned laser tracks reveal three distinct microstructural regimes that can be controlled by adjusting laser-scan parameters: columnar, dendritic, and banded microstructures. These regimes are consistent with those predicted by the analytic solidification theory for conduction-mode welding, but not for keyhole-mode tracks. The results provide insight into the fundamental laser–material interactions of the WE43 alloy under AM-processing conditions and are critical for the successful implementation of LPBF-produced WE43 parts in biomedical applications.

**Keywords:** additive manufacturing; magnesium; laser powder bed fusion; X-ray imaging; electron microscopy; microstructure

## 1. Introduction

The high strength-to-weight ratio and bioresorbability of magnesium-based alloy make it attractive for temporary implant applications [1], in particular for osteosynthesis. However, processing challenges and inhomogeneous corrosion may lead to early implant failure, which has prevented Mg-based alloys from being widely adopted [2]. The main factors controlling the corrosion rate and homogeneity of the corrosion front are the grain size and spatial distribution of the alloying elements within the microstructure [3–11]. Control of the microstructure, and therefore corrosion front homogeneity, has proven difficult with

traditional processing. Mg-based alloys are conventionally processed using melt casting, which does not allow fine geometry control or complex geometries [12,13]. Furthermore, melt casting often results in a coarse microstructure with large intermetallic precipitates due to slow cooling rates [12,13]. Rapid quenching techniques have been explored to refine the microstructure and delay secondary-phase precipitation by trapping solute atoms in the Mg matrix [4,8,11,14]. However, for structural applications, secondary processing is often necessary, which can result in the decomposition and coarsening of the non-equilibrium as-quenched microstructure [15]. Extrusion and other severe plastic deformation techniques have been applied to reduce grain size, but these methods inherently result in heavy mechanical deformation [16–18]. Spark plasma sintering has recently been proven to provide a refined grain structure but lacks the geometry control to achieve complex structures [15,19,20].

Laser powder bed fusion (LPBF) additive manufacturing (AM) is an attractive processing method as it offers both the rapid cooling rates necessary for microstructural refinement and the geometrical control needed for complex component geometries [21,22]. Recent work focused on the columnar to equiaxed microstructural transition in steel by LPBF and revealed that the modification of local thermal gradients by laser beam shaping allows tailoring of the local microstructure [23,24]. LPBF also offers great flexibility in implant geometry, allowing customization tailored to a specific patient. However, applying LPBF to Mg-based alloys presents many unique challenges. The first one is the high reactivity of Mg, which necessitates stringent atmosphere control measures to avoid the oxidation of the liquid melt and newly solidified weld bead. To minimize the danger during handling and shipping, the Mg powder particles are usually passivated with an oxide layer, which may be incorporated into the LPBF-processed material. The second challenge is the relatively small temperature difference between the melting (645 °C) and vaporization (1107 °C) temperatures of Mg [25]. The density of metal vapor in the plume produced during LPBF of Mg-based alloys is therefore expected to be higher than that of most commonly used AM metals. This can result in the obstruction of the laser beam and in compositional changes in the fabricated part compared to the powder feedstock due to the selective evaporation of Mg [26–28].

Single-track LPBF experiments are ideally suited to study the fundamental impact of varying process parameters on the laser–matter interaction and initial material response. Previous single-laser-track investigations of pure Mg have shown that an increase in laser-energy density increases grain size and reduces microhardness [29]. Further studies focused on the corrosion behavior of pure Mg material processed using several different laser parameters and revealed that the presence of defects, such as lack-of-fusion pores, leads to a lowered corrosion resistance [30]. However, LPBF processing of the Mg–Zn–Zr alloy ZK60 has shown promise, with improved mechanical and corrosion properties resulting from a refined microstructure after processing [31]. While LPBF processing of WE43 has been demonstrated [26,27], the study of laser–material interaction dynamics during LPBF of Mg-based alloys is limited to date. The process parameter window studied in the existing literature is narrow, because most studies use a combination of single-laser-power and scan speed, and several of them rely on broadening the laser beam up to approximately 125 μm to prevent Mg vaporization [25–27,32], which inherently limits the minimum printable feature size.

In addition, single-track LPBF experiments allow the use of more advanced in situ diagnostics, such as real-time X-ray imaging, which provides insights into the vapor depressions that form due to material vaporization, and the subsequent recoil pressure that drives the melt-pool surface deeper into the material. The method relies on the significant density contrast between both the solid or liquid and the vapor phase present in the depression. These experiments can yield not only the maximum depression depth achieved during the LPBF process, but also the dynamics of its depth and shape, as well as the circumstances of the subsequent pore formation and motion in the melt and during solidification. The method has recently been used in combination with modeling to reveal

the complex vapor recoil and melt dynamics that drive defect formation and material ejection in LPBF for a variety of materials [28,33–35], including stainless steel, Inconel, aluminum, and titanium alloys. However, studies on Mg-based alloys have not yet been reported, even though X-ray imaging is particularly attractive here, because of the high vapor pressure of Mg that drives sub-surface dynamics.

WE43, a commercially available Mg alloy with rare earth (RE) elements, has been established as one of the state-of-the-art metallic biodegradable implant materials due to its good mechanical performance and slower corrosion rate compared to commercially pure Mg [2,36–40]. Previous reports of LPBF of WE43 alloy provide information on the microstructure, corrosion, and material composition, but under rather limited laser processing regimes. Gieseke et al. [27] attempted to address the selective evaporation of Mg by pressurizing the process chamber to 0.3 bar above atmosphere to increase the Mg vaporization point [41]. Laser modification of an as-cast surface resulted in a refined microstructure, and slower, more controlled corrosion behavior [42]. Under relatively large laser-beam-size conditions (125 μm diameter, 700 mm/s, 200 W), improved microstructure and mechanical properties were realized in LPBF-processed material compared to material processed by powder extrusion and melt casting [27]. Several studies have specifically shown the potential for LPBF of WE43 to produce scaffold structures with complex geometries and interconnected pore structures, which may be used as orthopedic implants. Jauer et al. [25] showed the effectiveness of increasing inert-gas flow to compensate for the excessive Mg evaporation, increasing final part density. A plasma electrolytic oxidation surface treatment has been shown to smooth the surface of a printed part, thus improving mechanical and corrosion properties [26]. The mechanical and biocorrosion behavior of a scaffold have been investigated and two distinct microstructural regimes have been identified at different locations in the part. An extensive microstructural analysis of LPBF-produced Mg found distinct microstructural regimes in the bulk and near the surface, revealing the effect of thermal cycling on the final part microstructure. As such, the microstructural features of this last layer are well represented by a single-track experiment [32].

In the present work, we investigate the laser–material interaction dynamics of WE43 using a combination of in situ X-ray radiography, metallography, and microstructural analysis via electron microscopy. We complement our experimental results with thermodynamical modeling of the potential precipitation phases as well as kinetics simulation of the microstructural morphology. A broad range of LPBF-relevant laser melting conditions are considered, examining the impact of a change in laser power, scan speed, and beam size on the shape and size of the vapor depression, melt-pool geometry, likelihood of pore formation, and microstructure of single tracks in WE43. Fundamental aspects of the impact of these parameters on LPBF-produced WE43 are discussed, and indications of the optimization of LPBF conditions are presented.

## 2. Materials and Methods

### 2.1. Powder Material

The WE43 powder used in this study had a composition of 4% Y, 3% Nd, 1% RE, 0.4% Zr, with Mg balance (weight %) and a particle-size range of 20–45 μm (Magnesium Elektron). The powder was formed by gas atomization and each particle had an approximately 20 nm thick passivation oxide layer comprised of MgO and $Y_2O_3$.

### 2.2. Laser Irradiation for Ex Situ Studies

Single-track LPBF experiments were performed under various combinations of laser power and speed using a custom test apparatus [43–45]. The single-track laser irradiation system consisted of a 1070 nm, 600 W fiber laser (JK Lasers, model JK600FL, Rugby, UK), which was directed through a 3-axis galvanometer scanner system (Nutfield Technologies). The focused laser beam was a circular Gaussian at the sample plate with D4σ diameters of 50 and 80 μm. The beam passed into the chamber through a high-purity, antireflective-coated fused silica window. The vacuum chamber was evacuated using a turbomolecular

pump and purged with argon while taking care not to disturb the powder layer. A chamber pressure of ~700 Torr (0.93 bar) was maintained for all experiments and was controlled using a combination of the argon purge and pumping rates. The gas inlet for the argon gas flow was positioned below the powder layer. The residual oxygen concentration at the beginning of experiments was <50 ppm. Oxygen content was continuously monitored throughout the experiment using a vacuum-compatible oxygen sensor (Zirox GmbH, Model XS22, Greifswald, Germany).

Laser parameters were chosen to cover those used in the literature and to explore beyond them (see Table 1). Each laser track was a minimum of 10 mm long to ensure a steady-state scanning regime in the center of the track. Tracks were irradiated using a custom LabVIEW-based control software to control the laser and galvanometer scanner system. The substrate consisted of a 25.4 mm diameter, 3 mm thick disc of rolled WE43 (Magnesium Elektron) cut by wire electrical discharge machining (EDM). The parts were then ground with P-600 SiC paper to remove the EDM-induced oxide layer and ensure that only the metal alloy remained in contact with the powder. A 50 μm powder layer was manually spread on top of the substrate with a steel razor blade to mimic the powder-spreading process in a conventional LPBF system. The thickness of the powder layer was controlled by ensuring that the substrate plate was 50 μm below the holder surface during spreading.

**Table 1.** Laser-scanning parameter matrix. Cells highlighted in gray indicate laser power–scan speed combinations that were applied in this study.

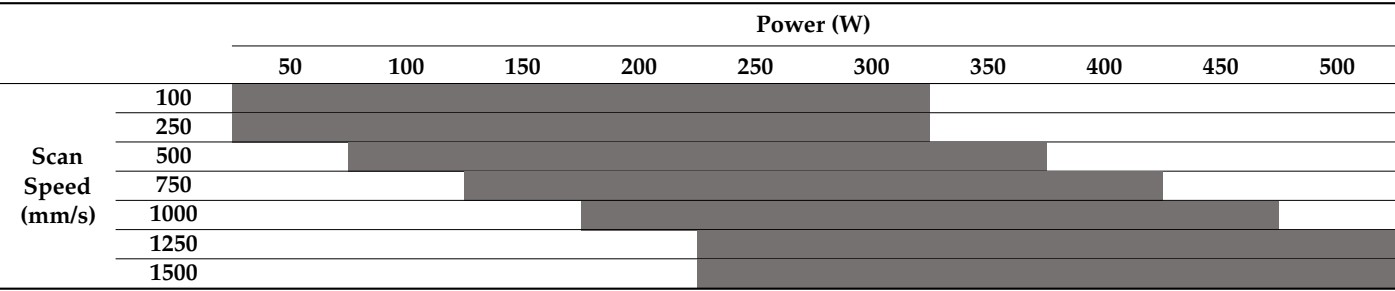

| | | Power (W) | | | | | | | | | |
|---|---|---|---|---|---|---|---|---|---|---|---|
| | | 50 | 100 | 150 | 200 | 250 | 300 | 350 | 400 | 450 | 500 |
| **Scan Speed (mm/s)** | 100 | ■ | ■ | ■ | ■ | ■ | ■ | | | | |
| | 250 | ■ | ■ | ■ | ■ | ■ | ■ | | | | |
| | 500 | | ■ | ■ | ■ | ■ | ■ | ■ | | | |
| | 750 | | | ■ | ■ | ■ | ■ | ■ | ■ | | |
| | 1000 | | | | ■ | ■ | ■ | ■ | ■ | ■ | |
| | 1250 | | | | | ■ | ■ | ■ | ■ | ■ | ■ |
| | 1500 | | | | | ■ | ■ | ■ | ■ | ■ | ■ |

### 2.3. Sample Preparation for Microscopy Analysis

Laser tracks were cross-sectioned using a wire diamond saw at least 2 mm from either end of the track to ensure a representative steady-state scanning section. Samples were prepared using standard metallographic techniques with ethanol as a lubricant instead of water to avoid oxidation during mechanical polishing. This surface finish was used for scanning electron microscope (SEM) imaging, whereas for optical microscopy, a solution made of 1 mL of $NHO_3$, 24 mL of $H_2O$ and 75 mL of ethylene glycol was used as an etchant, immersed in 5 s intervals, to reveal microstructural details. Specimens for electron backscattered diffraction (EBSD) were subjected to $Ar^+$ broad ion-beam milling (Hitachi IM4000, Chiyoda, Japan) as a final step. The broad ion-beam milling was performed in two steps. An initial short-duration ion-beam etching removed the top damaged layer resulting from preparatory grinding, and a subsequent long-duration polishing was performed at low accelerating voltage (2 kV) under a grazing-incidence angle. Transmission electron microscope (TEM) samples were prepared using an FEI NanoLab600i DualBeam focused ion beam (FIB) instrument to mill and lift out approximately 300 nm thick electron-transparent lamellae. The lamellae were extracted from the center of the track near the top surface.

## 2.4. Microscopy

Light microscopy of track cross-sections at each laser-processing parameter was performed using an Olympus microscope (DSX-CB). The depth and width of each track were measured and the molten and heat-affected zones were identified. Image analysis, including track-dimension measurements, was performed using the ImageJ software package [46]. A Hitachi SU70 SEM was used in both the secondary electron (SE) and backscattered electron (BSE) modes to extract the microstructure and alloying element distribution of powder particles and track cross-sections. Scanning TEM (STEM) imaging and energy-dispersive X-ray spectroscopy (EDS) chemical mapping were performed at ScopeM ETH Zurich using an FEI Company Talos F200X equipped with a 200 kV field-emission gun (FEG) and a 'SuperX'-EDS large collection-angle detector (Bruker). The chemical information obtained by EDS mapping in STEM mode was quantified using the ESPRIT software package of Bruker with the Cliff–Lorimer method.

EBSD was utilized to characterize the grain size and orientation distribution within the microstructure. EBSD was performed using an FEI Quanta 200 FEG-SEM, equipped with a Hikari Super CCD camera, operated at an accelerating voltage of 30 kV. The specimen was tilted to an angle of 70° for measurement. A step size of 100 nm and a hexagonal grid type was implemented. A binning of 4 × 4 was used for the CCD camera when capturing the Kikuchi patterns, producing 320 × 240 resolution images. The acquired raw EBSD data were subsequently analyzed using EDAX-TSL OIM™ Analysis 8 software. In order to achieve high indexing rates, the raw unfiltered EBSD data were reindexed using a neighborhood-pattern averaging scheme (n-PAR), which improves the signal-to-noise of the ratio patterns [47]. The n-PAR algorithm is available as a built-in function in the conventional EBSD analysis software OIM™ Analysis 8. Noise filtering was performed with a threshold confidence interval of 0.2.

## 2.5. In Situ X-ray Imaging

X-ray radiography was conducted at the Stanford Synchrotron Radiation Lightsource (SSRL) bending magnet beamline 2-2 using the LPBF testbed chamber described in detail elsewhere [35] and a scintillator-based high-speed X-ray imaging system [33]. Briefly, the LPBF system consisted of a 1070 nm continuous wave (CW) Yb-fiber laser (500 W maximum power, YLR-500-WC-Y14, IPG Photonics) coupled to a galvanometer scanning mirror system (Nutfield Technology, 3XB 3-Axis Scan Head). The laser was focused to a D4σ diameter of approximately 50 μm at the sample plane for all experiments. The vacuum chamber containing the sample was first evacuated to $5 \times 10^{-2}$ Torr and then filled with argon to 730 Torr (0.97 bar). Argon was constantly flowed through the vacuum chamber during experiments at 500 standard cubic centimeters per minute (SCCM) just above the powder layer. During processing, the laser was scanned using various laser-power and scan-speed conditions onto the thin edge of a 500 μm wide sheet of WE43, which was machined and polished with the same procedure as described above with the single-track samples. The WE43 substrate was sandwiched between two 1 mm thick glassy carbon sheets, which provided a trench to contain an approximately 50 μm thick manually spread WE43 powder layer on the substrate surface.

In situ X-ray images used the unfiltered white-beam X-ray spectrum of the SSRL beamline 2-2. Transmission X-ray images were captured using a scintillator-based optical system. The imaging system comprised an X-ray shutter (Uniblitz), 100 μm thick YAG:Ce scintillator crystal (Crytur), Ag-coated turning mirror (Thorlabs), 10× long working distance infinity-corrected objective lens (0.28 NA, Mitutoyo), tube lens (Thorlabs), and a FASTCAM SA-X2 1080K high-speed camera (Photron). This imaging system yields an effective pixel size of 2 μm for all X-ray images. Images were captured with a field of view of 1024 × 672 pixels at 20 kHz and an exposure time of 25 μs. X-ray images were analyzed using ImageJ [46] and Mathematica (Version 11.1.1) [48] software packages. Time-difference X-ray images were produced through the division of the uncorrected time-resolved image by the initial pristine substrate image. The resulting corrected radiographs are 2D projec-

tions, normal to the path of the laser beam, where the contrast indicates a change in density vs. the initial frame before melting. Lower-density features, such as pores and the vapor depression, presented dark contrast, whereas areas where more material was added, such as moving powder ejected by the beam or the track bead, showed light contrast. A custom script in Mathematica utilized the built-in Binarize contrast threshold method to identify and characterize pores in the irradiated WE43 substrate. To investigate the potential effect of thermal boundary conditions on the temperature profiles present in these experiments, we used the same finite-element analysis of laser heating and thermal transport simulation as those used in Calta et al. [35], but for the WE43 thermal properties [49,50].

### 2.6. Thermodynamic and Kinetic Modeling

The computational thermodynamic analysis of WE43 was performed based on the CALPHAD (CALculation of PHAse Diagrams) methodology [51], using the TCMG5 thermodynamic database of the Thermo-Calc software [52] for Mg-based alloys. WE43 was analyzed as 91.6% Mg, 4% Y, 3% Nd, 1% RE, 0.4% Zr (wt.%), where the 1% of RE elements were approximated as Nd. The total amount of Nd used for the analysis was therefore 4 wt.%. Predictions for the Mg–Y–Nd–Zr system were based on evaluations of the underlying binary and ternary systems; the 6 binaries except Nd–Zr and 1 of the 4 ternaries (Mg–Y–Nd) were available for the analysis. By including Y, we assumed that the majority of Y was solubilized, i.e., it was not trapped in the $Y_2O_3$ phase ($T_{melt}$ = 2425 °C).

A kinetic analysis of solidification morphology was performed using the Kurz–Giovanola–Trivedi (KGT) theory of directional solidification [53]. This theory predicts whether the morphology of a binary alloy is planar, cellular, or dendritic as a function of the thermal gradient $G$ and the solidification velocity $R$. WE43 must be reduced to an effective or pseudo-binary alloy for the analysis. We used the mass-weighted averages of the thermodynamic and kinetic properties of Y, Nd, and Zr to generate the effective binary parameters needed. These were based on the solute mass fraction $\bar{c}$, partition coefficient $k$, liquidus slope $m$, Gibbs–Thomson coefficient $\Gamma$, and solute diffusivity in the liquid phase $D^l$. We analyzed both limiting cases of Y behavior (fully melted/solubilized and fully unmelted/oxidized) by including or excluding Y from the averages.

A subscript notation $X_{i,j,k}$ was used to specify the effective solute composition, where $X$ was a material parameter and $i, j, k$ were the mass fractions of Y, Nd, and Zr considered, respectively, relative to their nominal values. For example, $\bar{c}_{i,j,k} = \sum_{n=i,j,k} nc_n$ where $c_n$ is the nominal mass fraction of element $n$. The solute mass fractions with 4% Nd, 4% Y, 0.4% Zr and 4% Nd, 0% Y, 0.4% Zr are thus $\bar{c}_{1,1,1} = 0.084$ and $\bar{c}_{1,0,1} = 0.044$, respectively. Other effective binary parameters correspondingly defined are $k_{i,j,k} = \sum_{n=i,j,k} nc_n^s / \sum_{n=i,j,k} nc_n^l$,

$m_{i,j,k} = -\Delta T / \left[ \sum_{n=i,j,k} n(\Delta T/m_n)^2 \right]^{1/2}$, and $D^l_{i,j,k} = \sum_{n=i,j,k} nc_n D^l_n / \sum_{n=i,j,k} nc_n$. Here, $c^s$ and $c^l$ are the equilibrium interfacial mass fractions, $\Delta T$ is the temperature interval from $T_1$ to $T_2$ over which $m_n$ is defined, $m_n = -\Delta T / \left[ c_n^l(T_2) - c_n^l(T_1) \right]$ is the liquidus slope for element $n$, and $D^l_n$ is the diffusivity of element $n$ in the liquid phase. The $D^l_n$ were computed using the phenomenological model of reference [54]. $\Gamma = \sigma T_m / L$ was taken to be solute-independent and the interface energy $\sigma$ was estimated from the negentropic model [55] as $\sigma = \alpha T L / T_m N_A^{1/3} V_m^{2/3} \approx 0.12 \, \text{J/m}^2$ at 620 °C. Furthermore, $T_m$ was the melting temperature, $L$ = 6700 J/mol was the latent heat, $\alpha$ = 0.86 was a geometric factor, $N_A$ was Avogadro's number, and $V_m = 1.35 \times 10^{-5} \, \text{m}^3/\text{mol}$ was the molar volume. We note that MD simulations of pure Mg [56] predict ed$\sigma \approx 0.09 \, \text{J/m}^2$ at 650 °C, comparable to our estimate for WE43. The final effective binary values obtained were $k_{1,1,1}$ = 0.23, $k_{1,0,1}$ = 0.12, $m_{1,1,1} = m_{1,0,1} = -18.9 \, \text{K/wt.\%}$, $\Gamma$ =2.2 × $10^{-7}$ K·m, $D^l_{1,1,1}$(828 K) ≈ $D^l_{1,0,1}$(828 K) = 2.4 × $10^{-9} \, \text{m}^2/\text{s}$.

## 3. Results and Discussion

### 3.1. Powder Characterization

The WE43 gas atomized powder particles generally present a spherical shape with a high surface roughness and smaller satellite particles, which adhere to the surface of the larger particles (Figure 1a). The microstructure of each particle consists of Mg-rich equiaxed hexagonal dendritic grains (Figure 1b). A secondary phase exists in the interdendritic regions, rich with heavier alloying elements, as indicated by the lighter contrast in the BSE image (Figure 1b). This interdendritic area is most likely a ternary $Mg_{14}Nd_2Y$ phase surrounded by the Mg matrix and enriched in Nd and Y (Figure 1c). In some cases, satellite particles appear to be embedded in the larger particle. The surface of the powder particles is covered by a mix of MgO and $Y_2O_3$, which offers good resistance to oxidation for the safe handling of the powder.

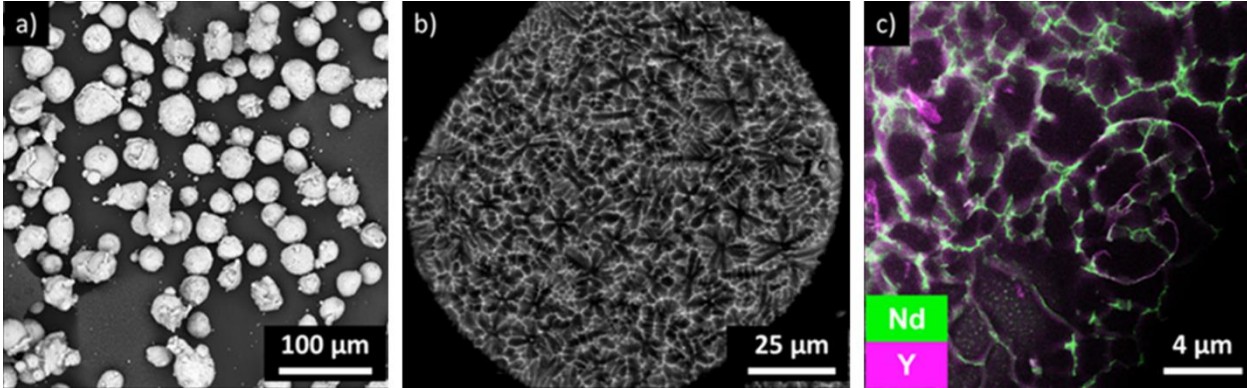

**Figure 1.** (**a**) SEM BSE image of WE43 powder revealing a 20–45 µm size distribution and spherical shape. (**b**) SEM BSE image of a cross-sectioned WE43 powder particle showing the microstructure: Mg-rich hexagonal dendrite structures are surrounded by a rare earth rich interdendritic phase. (**c**) STEM EDS elemental map revealing the distributions of Y and Nd in the powder microstructure.

### 3.2. Vapor Depression Behavior and Geometry

Figure 2 presents X-ray images captured during laser melting with a laser power of 150 W at a scan speed of 100 mm/s. Similar images were recorded for each laser power and scan speed of the parameter matrix presented in Table 1. In these videos, the laser comes vertically from the top and scans from left to right across the substrate. Note that because of the high energy density at low speeds and high powers, the melt pool was sufficiently wide, so that the glassy carbon windows influenced the melt-pool behavior by changing the thermal boundary conditions of the melt. To understand the influence of these thermal boundary conditions, simulations were performed to evaluate this effect on the melt temperatures. The glassy carbon indeed was shown to increase the subsurface melt temperatures, but only at low scan speeds (250 mm/s and below). The most significant influence was observed at 100 mm/s, while only minor changes were observed at 250 mm/s. At all higher speeds explored, the influence was negligible. Therefore, while the fluid flow in the example presented in Figure 2 was likely influenced by these thermal boundary conditions, the majority of the other conditions presented here were not affected. From these videos, several events were identified (Figure 2a):

1.  Movement of the powder particles ahead of the laser, visible as light contrast above the substrate surface (labeled 'ejected powder');
2.  A depression in the surface of the melt-pool due to recoil pressure, appearing as a narrow dark region that protrudes into the substrate following the laser scan (labelled 'vapor depression');
3.  The formation of pores, which appear as variously shaped dark contrast regions.

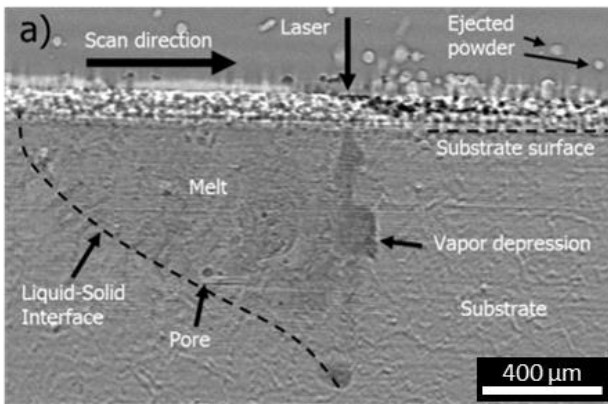

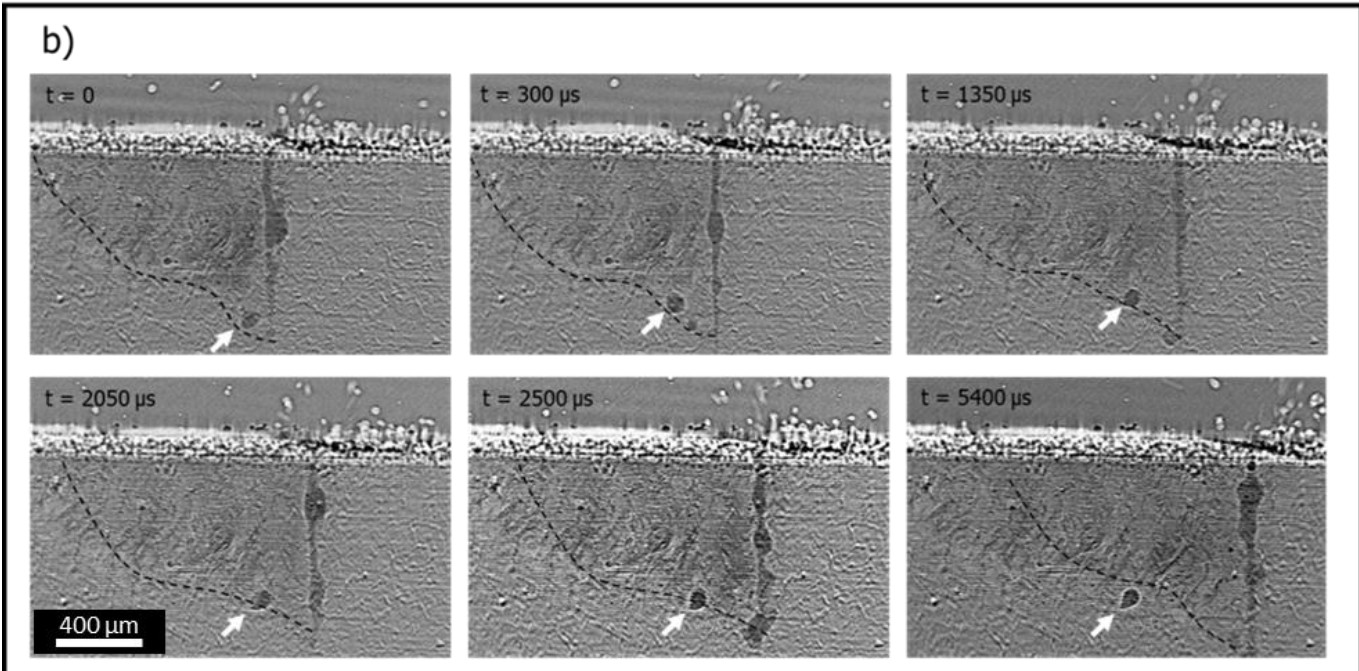

**Figure 2.** (**a**) In situ X-ray image of WE43 acquired during laser scanning revealing: the vapor depression, melt-pool boundary, formation of pores, and ejection of powder particles. (**b**) In situ X-ray images captured during the $P$ = 150 W, $v$ = 100 mm/s scan revealing the formation and solidification of one pore (white arrow). The pore originates at the base of the vapor depression, is buffeted around in the melt, and is captured and pulled toward the vapor depression by the solidification front, resulting in a balloon-shaped pore.

Figure 2b presents several frames captured during a 150 W, 100 mm/s irradiation, showing an example of pore formation. The full video is included in the Supplementary Materials, Video S1. In each frame, the vapor depression is visible as a vertical feature of darker contrast on the right side, the depth of which extends to the bottom edge of the field of view. Above the substrate surface, one can observe powder that has been ejected by the incoming laser. Although the contrast difference between the solid and the liquid is minor, the melt pool could be roughly outlined (represented by the black dashed line), thus revealing the liquid–solid interface. In the first frame, the pore in question can be observed just to the left of the vapor-depression base, having recently separated from the vapor depression. For the first 1350 μs, the pore follows a meandering path near the bottom of the melt, likely caught in an eddy current within the melt as a result of Marangoni flow [57]. A total of 2050 μs after the pore was created, the back of the pore appears to have been trapped by the solidification front, while the front of the pore is still in motion in the melt. The solidification front then passes through the pore, solidifying around it and pushing the

gas forward so that the back half of the pore appears narrow and pinched, while the front half is more rounded. After 5400 µs, the solidification front has passed the pore, which is frozen in the solid. The exact shape of this pore, which can strongly affect the stress concentration around it and initiate a crack under loading, is therefore a direct result of its interaction with the solidification front. This behavior is similar to observations by Martin et al. [35] in Al alloys and is likely material-independent behavior. Although the slow speed of the scan depicted in Figure 2 may not be relevant for LPBF processing, the slower solidification allows a more resolved picture of the influence of the solidification front on pore shape.

The morphology of the vapor depression can be divided into five distinct regimes, as presented in Figure 3. The depth ranges reported in this section are the minimum and maximum average values for the scans in each regime and show that some regimes have a wide range of depths while others show less variability. These vapor-depression shape regimes are therefore differentiated on the shape and dynamics of the vapor depression rather than the measured depth values. For the full videos of each regime, see Supplementary Materials, Videos S2–S6. At a low scan speed (100–250 mm/s) and power (50 W) (Figure 3b), the vapor depression was narrow, with a depth range of 73 to 137 µm. The average depth and shape of the vapor depression were relatively constant within this regime. With increasing power (100–300 W) at the lowest speeds (100–500 mm/s), the vapor depression remained narrow but was considerably deeper, with a depth range of 192 to 1017 µm, and presented large bulbous protrusions of vapor that originated at the base of the depression and traveled up to the surface with the moving laser (Figure 3c). These protrusions are similar to those previously observed in the Ti-6Al-4V alloy [33]. These protrusions not only extended to the back of the vapor depression, but also affected the front wall of the depression. The interface between the liquid melt and the vapor depression was unstable. This instability resulted in many pores forming behind the vapor depression. At the slowest scan speed and highest power, the vapor depression was deep enough to extend outside the field of view of the X-ray imaging experiments. With increasing speed (500–1250 mm/s) and power (150–350 W), see Figure 3d, the average depth of the vapor depression decreased, ranging from 194 to 752 µm. It did not show the same bulbous protrusions, but instead exhibited an oscillating bulge to the rear near the middle/base of the depression and a stable front wall. A small subset of the highest power (300–400 W) at intermediate speeds (750–1000 mm/s) presented an even larger bulge at the rear of the depression, which was located higher in the depression (Figure 3e). The average length of the vapor depression increased and the average depth also increased, ranging from 408 to 560 µm. At the highest speeds (1000–1500 mm/s) and powers (250–500 W), the vapor depression extended further and was shallower than in the other regimes, with a depth range of 194 to 539 µm (Figure 3f). At these high speeds, the vapor depression was highly dynamic and significantly changed shape compared to the other regimes, with a tail feature and a much longer vapor depression.

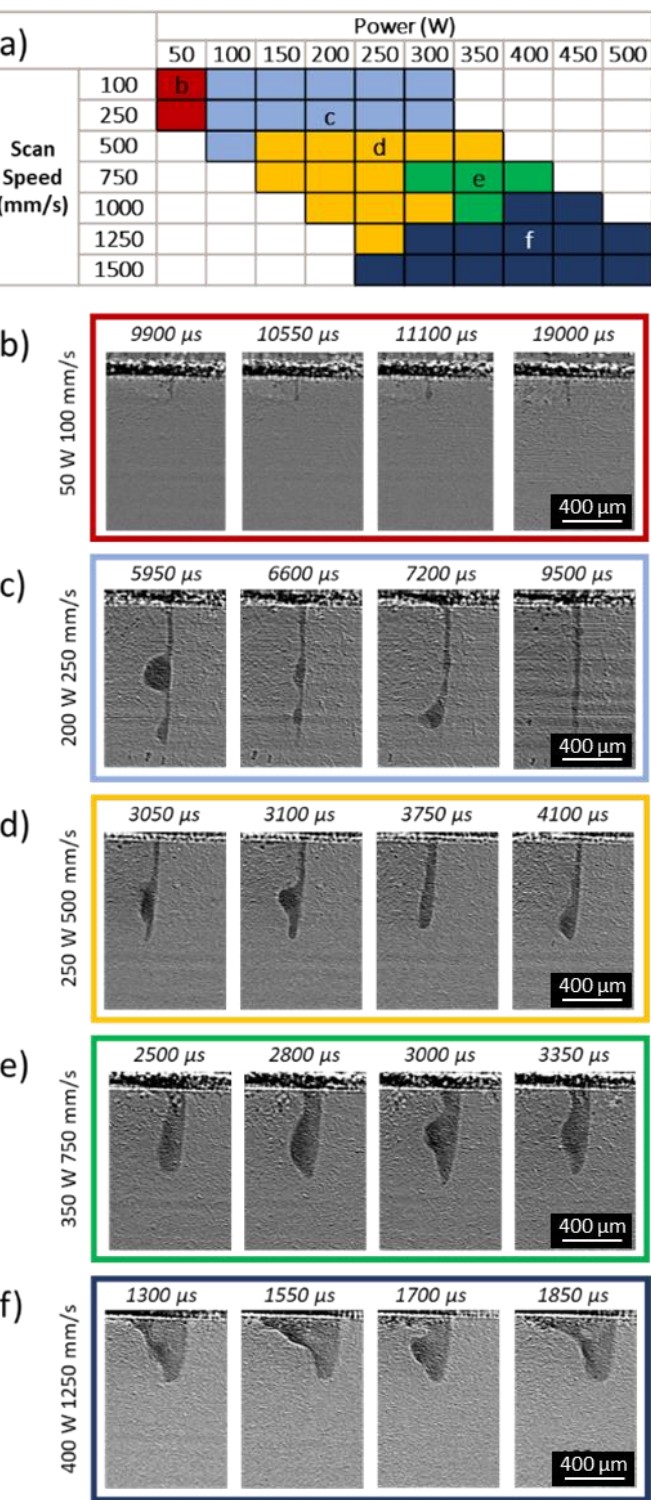

**Figure 3.** (**a**) Parameter matrix showing five regimes of vapor depression shape. (**b**–**f**) Representative in situ X-ray images for each regime: (**b**) *P* = 50 W, *v* = 100 mm/s, showing a thin, shallow vapor depression. (**c**) *P* = 200 W, *v* = 250mm/s, showing a deep thin vapor depression with large bulbous protrusions. (**d**) *P* = 250 W, *v* = 500 mm/s, showing a thin mid-depth vapor depression with a persistent oscillating protrusion to the rear. (**e**) *P* = 350 W, *v* = 750 mm/s, showing a thick mid-depth vapor depression with a persistent oscillating protrusion to the rear. (**f**) *P* = 400 W, *v* = 1250 mm/s, showing a thick mid-depth vapor depression with a trailing tail opening up to the surface.

*3.3. Pore Formation and Properties*

Figure 4 shows a typical analysis of such an X-ray in situ imaging of LPBF-processed WE43. Pores formed during the scanning process were identified in the final frame of the X-ray image sequence when the laser was off (pores indicated by white arrows in Figure 4a), and their depth and cross-sectional size were recorded as a function of laser parameters. Pores indicated in black are formed as a result of end-of-track phenomena and were not included in our pore count or in our further analysis, as they are not representative of steady-state behavior. In Figure 4b and c, pore depth and size are plotted as a function of normalized enthalpy, defined as:

$$\beta = \left( \frac{AP}{\pi H_m \sqrt{Dva^3}} \right) \tag{1}$$

where $a$ is the beam radius ($a = 2\sigma$), $A$ is the absorptivity, $P$ is the laser power, $v$ is the scan velocity, $H_m$ is the volumetric melting enthalpy, and $D$ is the thermal diffusivity, according to the normalized enthalpy relations based on [58] and described in detail below. Pores formed at the laser shut-off point (indicated by black arrows in Figure 4a) occurred for each condition and were not recorded in these data sets as they are characteristic of a transient behavior associated with the laser turning off rather than intrinsic steady-state behavior. An accurate pore count could not be made in tracks where the vapor depression extended outside the field of view, so these cases were also excluded. It is interesting to note that for speeds above 500 mm/s, pore formation was not observed during steady-state scanning (Figure 4c). Other than this speed threshold, the number of pores did not appear to depend on the process parameters. Instead, the threshold was likely dependent on the variation of the stability of the liquid–vapor interface with laser-energy density. The stability of the liquid–vapor interface is a balance between the liquid surface tension and the recoil pressure of the evaporating metallic vapor. This surface tension decreases with the increasing temperature. The laser conditions that produce the highest temperatures are not necessarily the highest laser power, but those with the highest energy densities. Previous reports in Al-based alloys showed that higher laser-energy densities result in the highest number density of pores [34]. The low vaporization point of Mg may result in a higher recoil pressure, which at high energy densities (and therefore low surface tension) is sufficient to drive vapor depressions deeper and lead to the collapse and instability of the liquid-vapor interface. Slower laser-scan speeds produced the highest energy densities and therefore the least stable interface with more opportunity for gas to be trapped by the collapsing liquid. As presented in Figure 3, the aspect ratio between the depth of the vapor depression from the surface and the width of the vapor depression in the laser-travel direction changes considerably with the laser-scan speed. For vapor depressions of similar depths, low scan-speed conditions produced high aspect-ratio depressions compared to those produced at higher scan speeds, even though the higher scan-speed conditions were produced with higher laser power. In the case of slower scan speeds and higher energy densities, any variation in laser absorption can produce the closure of the vapor depression and the entrainment of gaseous material as pores. This is supported by pores frequently forming at the base of the vapor depression, and the vapor depression depth increasing with increasing laser power. This phenomenon results in a linear relationship between increasing normalized enthalpy, $\beta$, and pore depth under conditions where $\beta$ is in the range of 0 to 100 (Figure 4b). When $\beta$ is increased above 100, pores are distributed throughout the depth of the melt pool and are larger in size, suggesting further instabilities at these higher energy densities (Figure 4c).

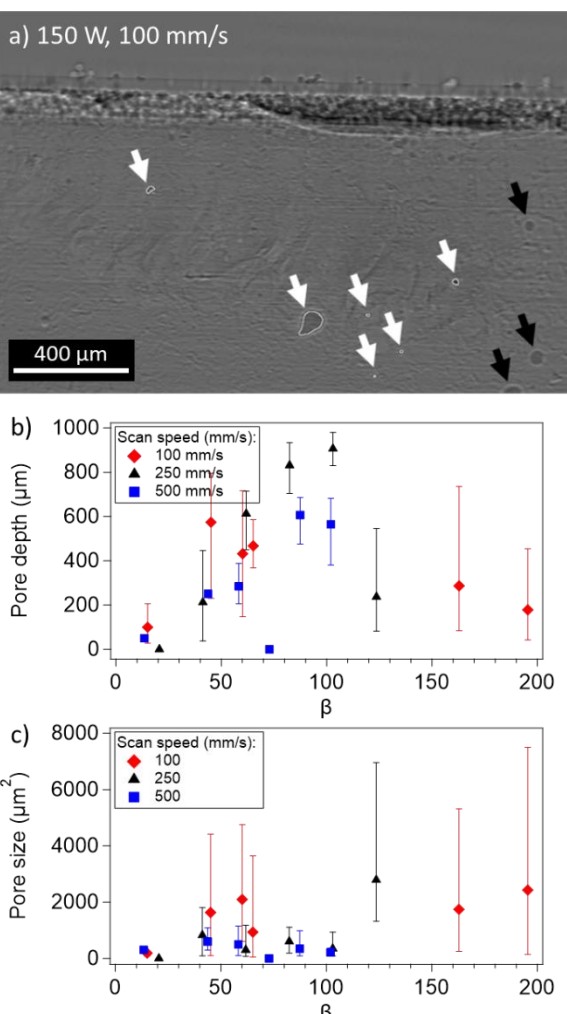

**Figure 4.** (**a**) Transmission X-ray image of WE43 showing pores formed during steady-state laser scanning (white arrows) and laser shut-off pores (black arrows to the right). The latter are not included in the pore count. (**b**) Average pore depth and (**c**) average cross-sectional pore size as a function of normalized enthalpy, *β*. Error bars indicate the minimum and maximum values. The beam diameter is 50 μm for these experiments.

### 3.4. Melt-pool Depth and Geometry

The ex situ analysis is presented in Figure 5. Light-microscopy images of cross-sectioned laser tracks on thick plates (as opposed to the thin substrates used for X-ray imaging) revealed the size and geometry of the melt pools for both ex situ sets. Track depths resulting from scans using 50 and 80 μm diameter beams are graphically reported in Figure 5a,b, with each point representing the melt-pool depth for a given laser condition (speed, power). The depth of tracks made with the 50 μm diameter beam at the highest speeds (>1000 mm/s) linearly increased with increasing power. Tracks made with lower speeds appeared to be more erratic, with fluctuations in depth obscuring any correlation with the process parameters. These fluctuations are especially visible for $v = 100$ mm/s, where the depth of the track ranges between 50 and 500 μm. For the tracks made with an 80 μm diameter beam, the depth increased linearly with power for each speed, but was on average much shallower than the depths of the 50 μm diameter beam set.

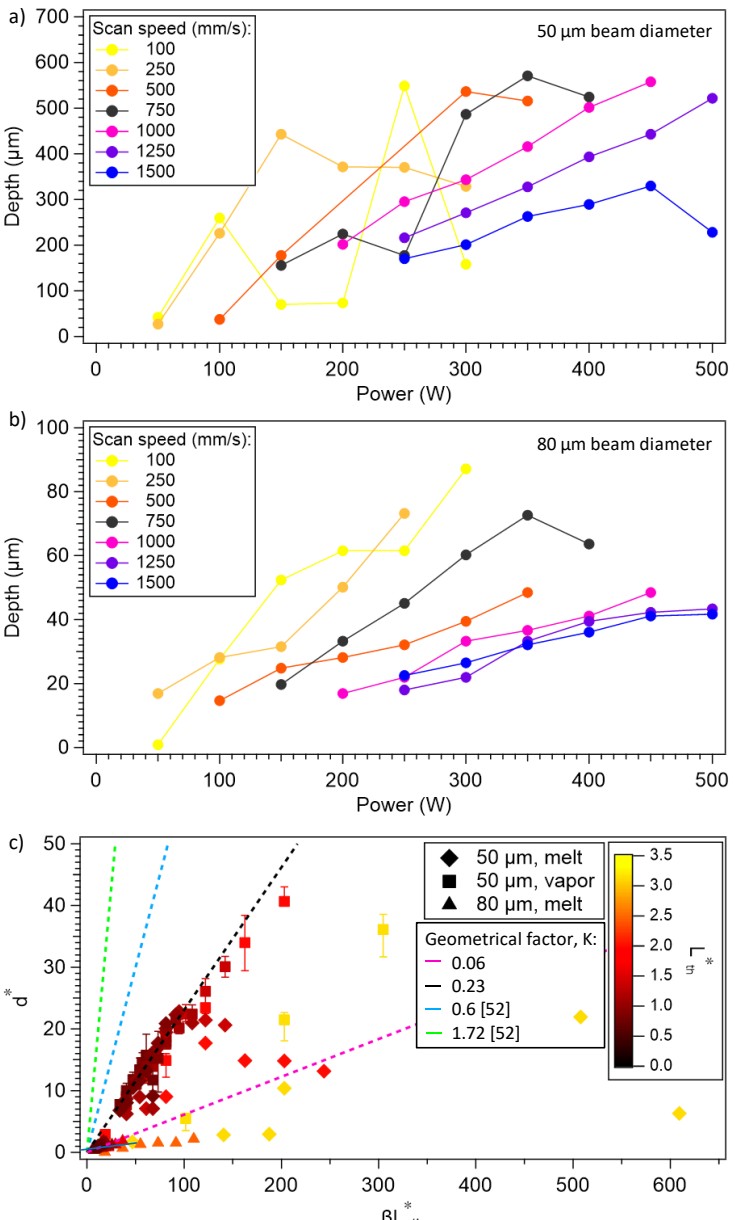

**Figure 5.** Analysis of the depth of the laser-induced melt pool in WE 43, derived from light microscopy performed on laser-track cross-sections. (**a,b**) Melt depth from the substrate interface for each scan condition measured ex situ using metallographic cross-sections. (**a**) Tracks scanned with a 50 μm diameter beam. (**b**) Tracks scanned with an 80 μm diameter beam. (**c**) Normalized depth ($d^*$, $d/a$) as a function of $\beta L^*th$ for WE43 with varying laser-scan conditions. The '50 μm, vapor' dataset is the vapor-depression depth measured using in situ X-ray imaging, and the error bars are the minimum and maximum values measured in the field of view. The geometric factor ($K$) is fitted to the '50 μm, vapor' dataset. Additionally, $K$ is shown for 316L stainless steel, Inconel 625, and Ti6Al4V in the conduction and keyhole-mode welding regime from [58]. Data points are colorized as a function of $L^*_{th}$.

Peak temperatures and therefore melting and vapor recoil depend on laser parameters, power, scan speed, and beam size. To help rationalize the variation in vapor-depression and/or melt-pool depth observed across these data sets, normalized enthalpy relations were computed and compared using the scaling laws derived in [58]. These scaling laws introduced a term for the thermal diffusion length, $L_{th}$, where $L_{th}=\sqrt{D/av}$. Figure 5c presents $d^*$ (depth, $d$, normalized by the beam size, $d^* = d/a$) plotted as a function of $\beta$

multiplied by the normalized thermal diffusion length, $L_{th}^*$ ($L_{th}^* = L_{th}/a$). The relationship between these terms is defined by:

$$d^* = \frac{d}{a} = \left( \frac{AP}{\pi H_m \sqrt{Dva^3}} \right) \left( \sqrt{\frac{D}{av}} \right) \tag{2}$$

$$d^* = K\beta L_{th}^* \tag{3}$$

where $K$ is a geometric factor approximately equal to 0.6 for conduction-mode welding in Inconel 625, 316L steel and Ti-6Al-4V [58]. Absorptivity during LPBF processing is known to considerably vary with laser parameters, ranging from ~0.3 to 0.8 in 316L steel [45]. In estimating the absorptivity of each state, we considered the value reported in the literature for solid WE43 [59] and the values measured experimentally for multiple materials transitioning from conduction to keyhole regimes [58]. Laser tracks in the conduction-mode welding regime, defined here as a melt depth of less than 150 μm, were assumed to have an absorptivity of 0.3, and deeper keyhole-mode tracks were assumed to have an absorptivity of 0.65. Figure 5c shows *d\** as a function of $\beta L_{th}^*$ for WE43 with varying laser-scanning conditions and beam diameters, with depths measured by ex situ melt track analysis or in situ X-ray imaging. Additionally, presented in Figure 5c are the best fit and *K* values for the conduction and keyhole-mode regimes in Inconel 625, 316L steel and Ti-6Al-4V from Ye et al. [58]. Note that the *K* value is nearly identical for all these materials. Although the 50 μm beam diameter data for WE43 is similar in magnitude ($K \approx 0.23$), the data from the 80 μm beam diameter condition diverges greatly from these other materials, with a value of 0.06. The divergence of WE43 from the other common LPBF materials can probably be attributed to excessive Mg evaporation caused by the narrow temperature difference between the melting and boiling points of Mg the higher thermal diffusivity of WE43. Comparing the thermal diffusivity of WE43 (0.243 cm$^2$/s) to Inconel 625, steel 316L, and Ti-6Al-4V (0.048, 0.050, and 0.086 cm$^2$/s, respectively) reveals considerable differences. A consequence of this increase in thermal diffusivity is the condition of $L_{th}^* > 1$. This results in transverse thermal transport becoming significant and therefore in the assumption that the normalized enthalpy scaling, where the absorbed laser energy is spent mainly to melt the alloy, does not hold anymore, contrary to the case of the other materials. Although this increase in thermal diffusivity makes the direct comparison between materials using normalized enthalpy difficult in this regime, it is interesting to note that the 50 μm beam diameter data collected using in situ X-ray imaging of the vapor pressure scales linearly even under the conditions where $L_{th}^* = 2$. Data where $L_{th}^* > 2$, for scans performed with high laser power at 100 and 250 mm/s, varied widely and was considerably outside typical LPBF conditions.

The data collected by the ex situ melt-pool analysis of the 80 μm beam diameter laser irradiations varied from the in situ X-ray data and were tentatively attributed to the differences between argon flow in the two chambers, which resulted in differences in the magnitude of laser–vapor interaction that scattered and distorted the build laser. This effect was expected to be particularly strong in Mg alloys, where the difference in temperature between the melting and equilibrium boiling points was quite small compared to other common LPBF alloys. The nonlinear depth trends at low speeds with a 50 μm beam diameter and slow scan speeds indicated a fluctuating transition between the conduction and keyhole regimes, which ultimately pointed to changes in the amount of laser energy absorbed in the melt pool, which we ascribed in this study to laser interaction with the vapor plume. Slow scan speeds resulted in higher temperatures, which not only produced a significant amount of Mg vapor, but since the direction of the vapor was normal to the sample surface and likely interfered with the incident laser through either scattering or absorption, this reduced the effective power on the sample surface [60]. In both cases, the reduced effective power on the sample surface would lead to a reduction in the melt depth and may even push the tracks from the keyhole regime to the conduction regime. This

effect is stochastic, resulting in depth fluctuations, as observed in Figure 5a, and is possibly a result of the inconsistent removal of Mg vapor in the build chamber.

### 3.5. Melt-Pool-Shape Regimes

In the cross-sectional geometry of the melt pools scanned with a 50 μm diameter beam, five shape regimes were identified (Figure 6). The cross-section shapes represented in Figure 6b–f were schematically derived from light-microscopy images of the laser conditions indicated with a corresponding letter in the table of Figure 6a. Note that the scale bar was relevant for all tracks. The lowest power (50–100 W) and speed (100–500 mm/s) tracks were shallow and narrow, with a minimal weld bead (Figure 6b). At low speeds (100–250 mm/s) and all but the lowest powers (100–300 W), the tracks varied in shape and geometry, but were generally deep and wide with a parabolic shape (Figure 6c). At moderate speeds (500–1250 mm/s) with higher powers (300–500 W), the geometry of the melt pool presented an unusual shape with a narrowing just below the surface and then a widening in the lower part of the track to resemble the shape of a bowling pin (Figure 6d). To the best of the authors' knowledge, this shape has not been reported before in LPBF scan tracks. At the lowest powers (150–250 W) and for all but the lowest speeds (500–1500 mm/s), the tracks presented small and relatively shallow melt geometries with a small bead (Figure 6e). Tracks scanned with an 80 μm diameter beam presented a geometry consistent with that observed in Figure 6b, but the depth was considerably deeper in some high-energy conditions. This type of analysis has been previously reported for stainless steel in the context of laser welding [61].

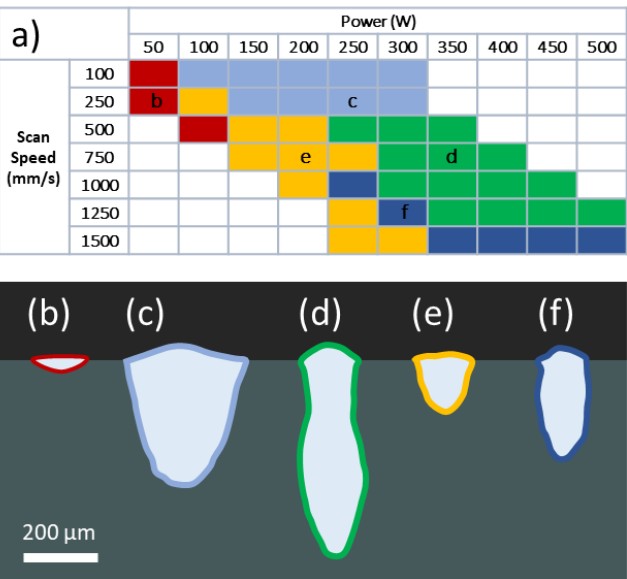

**Figure 6.** (**a**) Classification of melt-track geometries revealing the shape regimes. (**b**) Small, shallow melt pool, (**c**) wide and deep melt pool, (**d**) unusual bowling-pin-shaped melt pool, (**e**) the melt-pool regime most suitable for printing, and (**f**) slightly deeper and thin melt pool.

It should be noted that most of the observed track morphologies reported in Figure 6 are too deep or wide to be considered useful for printing complex geometries, but the track marked in yellow in Figure 6e has a melt-pool geometry suitable for LPBF. The bowling-pin shape observed in Figure 6d is unique, likely caused by excessive evaporation in the keyhole that pushed the melt outwards, followed by rapid solidification (Figure 3d). The tracks printed with an 80 μm diameter beam were shallower and all presented similar track shapes consistent with those in the 50 μm beam diameter tracks scanned with the slowest speed (Figure 6b). The limited parameter space in which a 50 μm diameter beam resulted in processible tracks compared to the 80 μm diameter beam was in agreement

with the use of a defocused or expanded beam in the literature to reach a reliable printing regime [25–27,32].

### 3.6. Microstructure Analysis

The microstructure within each track was probed using a multiscale microscopy approach. Tracks were imaged in cross-sections near the surface with SEM in BSE mode to observe the microstructure and the distribution of the alloying elements (Figure 7). Our laser melting experiments resulted in a very fine microstructure with most grains measuring below 10 μm in diameter, as compared to the 50–100 μm grain size achievable with melt casting [13]. A boundary area of not greater than 10 μm around the edge of the melt pool showed the heat-affected zone (HAZ), where the substrate microstructure was altered by the heat of the melt, but it was not melted. In this area, the grain size appeared to be larger than in the bulk of the substrate, but the secondary-phase precipitates did not appear visibly larger.

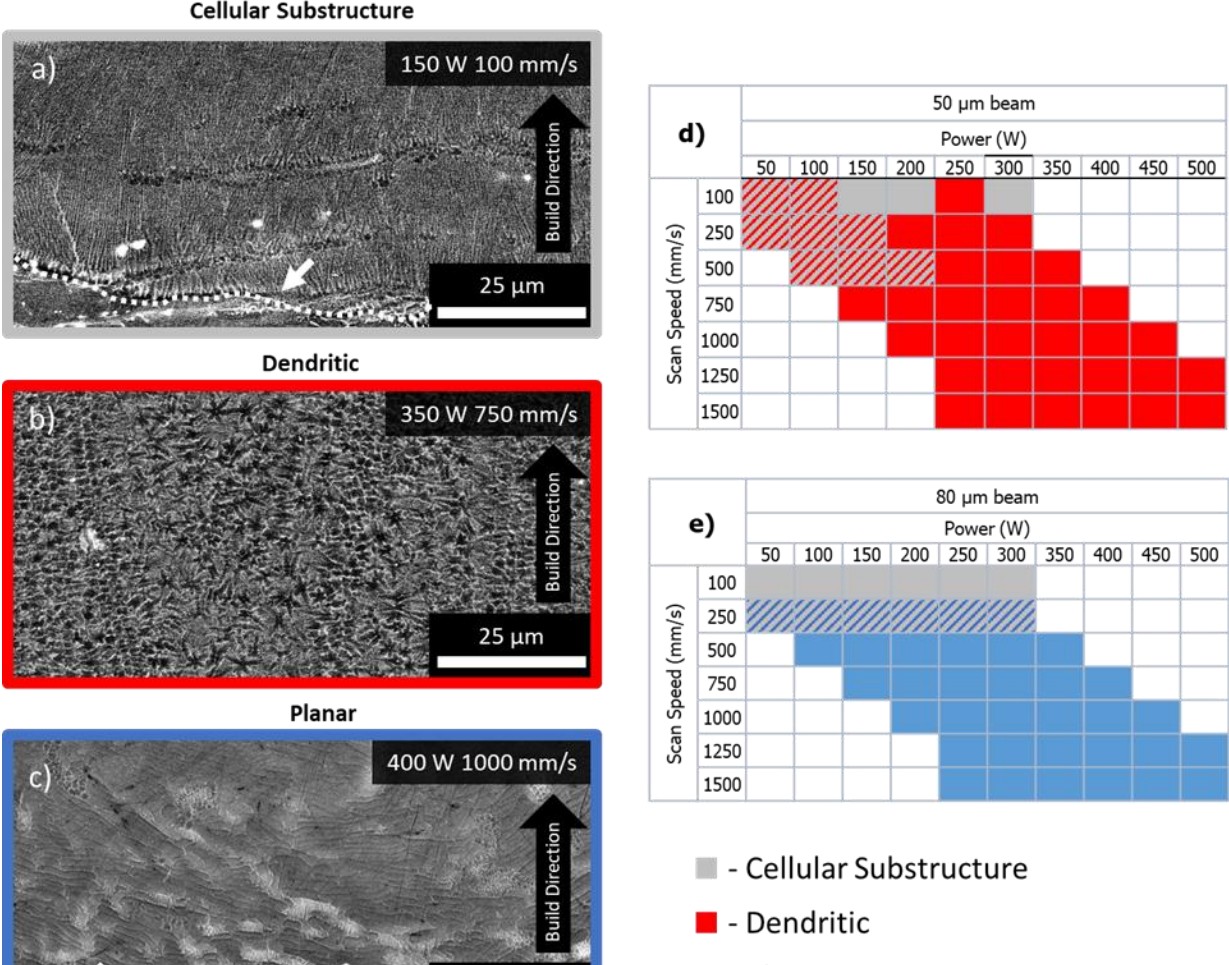

**Figure 7.** Examples of the three microstructural regimes observed within the melt pools in laser tracks in WE43. (**a**) The vertical cellular substructure observed in both 50 and 80 μm diameter beam-scan sets at the slowest speeds and high powers; the melt pool is outlined in white and indicated by a white arrow. (**b**) The equiaxed dendritic structure observed at high speeds in the 50 μm beam diameter set. (**c**) The planar banded structure observed at high speeds in the 80 μm diameter beam set. (**d**,**e**) Parameter matrix showing microstructure regimes of (**d**) tracks scanned with a 50 μm diameter beam and (**e**) tracks scanned with an 80 μm diameter beam.

Within the tracks, three distinct microstructural types were observed (Figure 7a–c). The first were grains with a fine cellular substructure (Figure 7a). These cellular substructures originated from the edge of the melt pool, growing away from the walls of the melt pool and toward the surface. The cellular substructures were measured to be approximately 5–10 μm in length and 300 nm wide. The center of each cell appeared dark in contrast and the intercellular area appeared lighter in contrast. This microstructure regime was observed in tracks scanned with both the 50 and 80 μm diameter beam size with the lowest speeds and mid-to-high powers. In many of the tracks that predominately showed the first type of microstructure of grains with a vertical cellular substructure, there were also lines of small equiaxed grains that ran parallel to the edges of the melt pool (Figure 7a). These may indicate an area of higher alloying-element content or thermal fluctuations due to convective flow leading to more rapid solidification. The vertical cellular substructure then continued to grow above these regions, with the same directionality as below, toward the surface. The second microstructure type, observed at a faster scan speed with a 50 μm diameter beam, was an equiaxed six-arm dendritic structure similar to those in the $P = 300$ W and $v = 750$ mm/s scans (Figure 7b). These dendrites were small, 1 to 3 μm in diameter. These grains were mainly oriented such that the basal plane was normal to the cross-sectional surface.

The center of the grain and the bulk of the arms appeared with dark contrast, whereas each of these grains was outlined in light contrast. Many scan parameters with the 50 μm beam diameter resulted in areas with both cellular and dendritic microstructures, the ratio of each varying with the scan parameters. At faster scan speeds with an 80 μm diameter beam, a banded microstructure was observed (Figure 7c). The bands were parallel to the melt-pool boundary and their thickness was similar to the cell size in the cellular type grains: ~300 nm. These three distinct microstructural regimes resulted from the different cooling rates induced by the varying scan parameters. The first microstructure type had cellular substructured grains that were oriented from the edge of the melt pool inward and upward to the center of the top surface. This was the opposite of the heat flow direction, indicating that the grains grew along the solidification front. This type of solidification is the most common for LPBF processes and has previously been observed in other materials [62]. These structures occurred at the slowest scan speed and highest-power laser conditions for both beam sizes. The six-armed dendritic grain shape observed in many of the tracks is congruent with the hexagonal Mg crystal structure. A nanoscale cluster of RE-rich material is often found at the core of these structures, which, having a higher freezing point, may have been the nucleation seed for the Mg grain. The directionality and presence of these RE-rich seeds indicates that these grains nucleated within the melt and did not solidify directionally at the melt-pool boundary. The Mg-rich arms are surrounded by the RE-rich interdendritic phase. This suggests that, as the grains grew, the RE elements remaining in the liquid phase were pushed to the interdendritic area. The equiaxed hexagonal dendritic grains are indicative of a high ratio between the temperature gradient $G$ and the solidification front velocity $R$ ($G/R$) compared to conventional processing conditions, such as melt casting, which is further supported by their small size. The banded microstructure observed in the tracks scanned with an 80 μm beam indicated an even higher $G/R$. These observations are discussed in detail below in the theoretical analysis section in the context of the WE43 solidification morphology map. At a sufficiently high solidification front velocity, the solidification was planar, perpendicular to the heat flow direction. This type of banded solidification requires high-solidification front velocities [63]. Near the surface of these tracks, there were often no discernible microstructural features, and this indicates that the solidification was truly planar, where the $\alpha$-Mg phase is growing in a steady state without nucleating any secondary phases. There was a general relationship between the geometry of the melt pool and the type of microstructure, with conduction-mode tracks showing more cellular and banded microstructures, and keyhole-mode tracks showing more equiaxed dendritic microstructures (Figure 7d,e). This illustrates that varying the processing parameters can influence the resulting microstructure in LPBF-produced WE43.

The arrangement of hexagonal dendritic grains in lines or swirls, similar to the structures observed in Figure 7a, was visible in many tracks with the equiaxed dendritic and banded microstructure types. The source of the swirls could be the result of either thermal or chemical fluctuations. Thermal fluctuations result from Marangoni flow currents in the melt. In some places, these currents lead to cooler spots in the melt where the temperature may reach the solidification temperature of the secondary phases, seeding solidification of equiaxed grains.

All tracks from both the 50 and 80 μm scan sets showed small (5–10 μm) spherical secondary-phase particles of bright contrast, indicating a high concentration of alloying elements (Figure 7a). These equiaxed RE-rich secondary-phase particles present in most tracks resembled particles of the same shape and size as those present in the substrate material. It is possible that these particles, which have a higher melting point due to their RE content, were not melted during the laser processing and were incorporated into the solidified microstructure. In the cases where the matrix surrounding these particles presented a depletion of alloying elements, evidenced by the darker contrast, it may be that the particle received Y from the matrix. Common to all melt-pool cross-sections was a region of thin, elongated grains visible just at the melt-pool-substrate boundary, seemingly growing toward the center of the melt pool. In many tracks, pieces of oxide shells that were present on the surface of the powder particles appeared to have been embedded in the material. In some cases, these shells still retained part of the spherical shape of the powder particles.

EBSD data were collected at two orthogonal angles to investigate whether the observed grains were truly equiaxed or if they were elongated in the solidification direction (Figure 8). Similar data were also collected on the nonmelted substrate to compare the grain sizes. We observed that grains were not elongated in the solidification direction and exhibited a basal fiber texture about the sample [1] axis, as defined in Figure 8, with the *c* axis perpendicular to the laser-scanning direction.

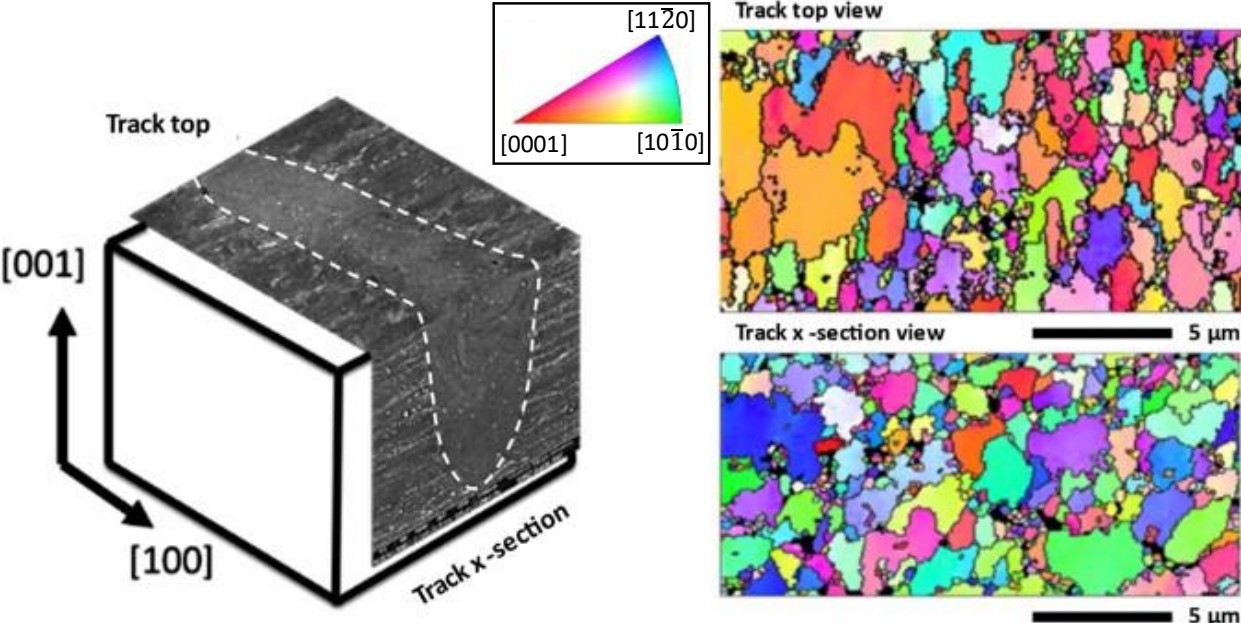

**Figure 8.** Electron backscatter diffraction (EBSD) data collected from the melted region of a cross-sectioned single track irradiated at *P* = 300 W, *v* = 750 mm/s. The left panel shows the orientation of the two orthogonal views with the solidified melt pool indicated by the dashed line. Top and cross-section EBSD images are presented to the right, illustrating an isotropic grain structure.

The microstructure regimes observed in this work may be compared to those previously observed on the top surface of full builds [32], where structures similar to both the equiaxed and banded microstructures observed in different areas of the track are reported. None of the three regimes defined here resembled the microstructure observed in the bulk because the cyclic annealing imposed by the addition of layers during the build caused a microstructural evolution throughout the process.

As many of the tracks in the parameter space were well outside of what was considered the printable regime, TEM investigations focused on the microstructures formed with the processing parameters most interesting for printing. Three samples produced with a 50 µm beam diameter were imaged with TEM and STEM EDS for a greater insight into the microstructure and its composition (Figure 9). Although the samples resulted from different laser parameters ($P = 100$ W, $v = 250$ mm/s; $P = 200$ W, $v = 750$ mm/s; $P = 300$ W, $v = 1500$ mm/s), each presented a track geometry that could potentially be useful for printing, as defined in Figure 6. The sample scanned at $P = 100$ W, $v = 250$ mm/s had a melt-pool depth of 200 µm (Figure 9a). This track presented a columnar structure with Mg-rich grains divided by areas rich in the alloying elements Y and Nd (Figure 9a,d), similar to the microstructure illustrated at longer length scales in Figure 7a. The matrix composition measured in this region was 93.80% Mg, 1.38% Y, 0.56% Nd, and 0.30% Zr (at.%). Small cuboid-shaped RE-rich particles were observed. They had an average composition of 54.5% Mg, 37.0% Y, 1.71% Nd, and 2.79% Zr (marked with [♦] in Figure 9a,c,e,g). At higher magnification, the EDS map revealed one such particle in Figure 9e. An area rich in Y and O that had an irregular shape and is composed of small cuboids (marked with [▲] in Figure 9a,c,d), whose composition and structure match that of the powder-particle oxide shells (Figure 9h), was also observed. Note that the exact composition measured in these features may have been influenced by the presence of the Mg matrix either above or below them, as they may not have extended the full thickness of the FIB lamella. As mentioned above, these features were observed throughout all tracks from each scan condition and were likely oxide-shell remnants from the passivated powder particles' surface that were incorporated into the track during the scan, but were not entirely dispersed.

The sample scanned at 200 W, 750 mm/s presented a similar track geometry with a depth of 230 µm (Figure 9b). The microstructure of this track was different from that of the first track. In this sample, the microstructure consisted of hexagonal dendritic structures where the arms of the dendrites were α-Mg and the interdendritic area was rich in RE elements. A feature at the top shows an area where the surface was interrupted, showing the Y-rich surface layer likely to be present at the entire track surface. The matrix composition of this sample was 89.60% Mg, 4.40% Y, 2.77% Nd, and 0.99% Zr (at.%) (Figure 9g).

The sample scanned at 300 W, 1500 mm/s presented a similar track geometry with a depth of 200 µm (Figure 9c). The microstructure of this track was similar to the sample melted at 200 W, 750 mm/s, presented in Figure 9b, with α-Mg dendrites and a RE-rich interdendritic area. The matrix composition of this sample was 95.60% Mg, 1.40% Y, 0.57% Nd, and 0.36% Zr (at.%). Two areas from this scan track (Figure 9e,f), represented in EDS maps, show examples of the hexagonal dendritic grains present in the samples scanned at both 200 W, 750 mm/s and 300 W, 1500 mm/s. Figure 9e shows a high-magnification example of the secondary-phase particles observed in all tracks. Figure 9f presents an example of the six-arm dendrite and, at the center, a small area rich in Nd and Y. This illustrates a case where an RE-rich area serves as a seed to nucleate a dendritic α-Mg grain.

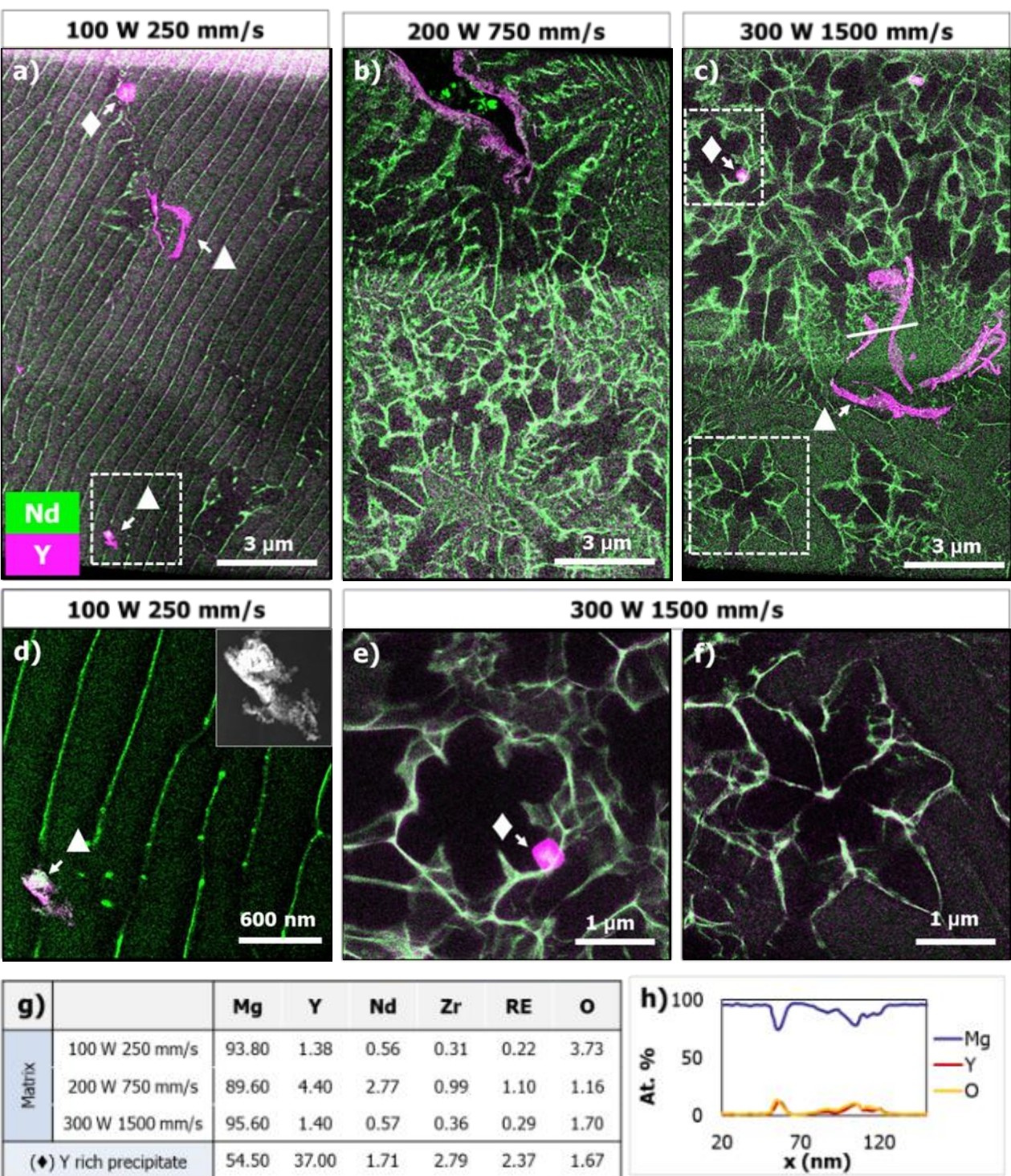

**Figure 9.** STEM EDS elemental maps in WE43 of three laser tracks with similar melt-pool geometries. Colorization of images shows the distributions of Nd and Y. (**a**) A 100 W, 250 mm/s sample showing a cellular structure; (**b**) 200 W, 750 mm/s sample showing a dendritic equiaxed structure; and (**c**) 300 W, 1500 mm/s sample showing a dendritic equiaxed structure. (**d**) Enlarged image from the highlighted box in (a) showing a cellular structure and Y-rich region; inset: magnified image of the feature denoted by ▲. (**e**) Enlarged image of a hexagonal dendrite structure with secondary-phase particle (♦). (**f**) Higher-magnification image of a hexagonal dendrite showing the RE-rich center. (**g**) Composition (at. %) of the matrix in (**a**–**c**) and the average measured composition of the secondary-phase particles (♦). (**h**) Composition of line scan (white line in (**c**)) across the powder-shell fragments showing the presence of both Y and O (▲).

The compositions of the matrix measured in the EDS maps from each scan track are presented in Figure 9g. Although there was a difference in the composition between the tracks, we observed no evidence of the systematic selective evaporation of Mg that correlated to the scanning parameters, and any differences were most likely due to local fluctuations in the composition. The TEM FIB lamella was small, relative to the size of the track, so the measured compositions were not necessarily representative of the overall composition throughout the melt pool. The loss of Mg through selective evaporation in a full build has been reported previously to correlate with scan parameters [64]. This effect was compounded in a full build where the material was scanned and molten repeatedly. Single tracks can be indicative of compositional differences, but the uncertainty is high due to the small sampling size.

### 3.7. Theoretical Analysis

The key aspects of our characterized microstructures can be understood in terms of thermodynamic predictions, presented in Figure 10, and classical theories of solidification kinetics. We first performed a computational thermodynamic analysis of WE43 based on the CALPHAD methodology [51]. The property diagram computed for equilibrium (molar phase fraction versus $T$; Figure 10a) predicted solidification into a single Mg-rich hexagonal close-packed (hcp) phase at ~637 °C, followed by the emergence of a small amount (initially ~3.29% by mole) of a ternary $(Mg)_{41}(Nd,Y)_5$ phase. The evolution of the liquid composition under equilibrium solidification conditions presents that the liquid at the solidification front is poor in Zr, rich in Y and rich in Nd (Figure 10b; Scheil nonequilibrium conditions produced similar results). The primary hcp phase is correspondingly Zr rich, Y poor, and Nd poor (but with more Y than Nd). We therefore expected solidification into the primary Mg-rich hcp phase with Y- and Nd-rich intercellular or interdendritic regions, consistent with the results of Figure 9.

We also performed an analysis of the solidification morphology using the Kurz–Giovanola–Trivedi (KGT) theory of directional solidification [53]. The computed $G$-$R$ morphology maps for compositions $\bar{c}_{1,1,1}$ and $\bar{c}_{1,0,1}$ are presented in Figure 11. The effect of excluding Y (the $\bar{c}_{1,0,1}$ case) is a modest increase in the size of the cellular/dendritic patterning regions at large $G$ and $R$. If we assume that $G \approx 2 \times 10^7$ K/m in the experiments and note that more detailed phase-field simulation studies [65] indicate that the topmost dendritic region predicted by the KGT theory tends to, in fact, be cellular (i.e., the topmost C-D boundary should be shifted downward), then the $\bar{c}_{1,0,1}$ predictions agree well with our observations for the 80 μm beam (Figure 7e). Specifically, we predicted and observed cellular morphologies between ~50–250 mm/s, and suppression of the cellular/dendritic instability above ~250 mm/s, in favor of planar or banded morphologies. A greater disagreement between the predictions and observations can be observed for the 50 μm beam (Figure 7d). Here, we predicted and observed cellular or partially cellular morphologies at low velocities ($R \approx 50$–250 mm/s), but the morphologies became more fine-grained and equiaxed-dendritic as $R$ increased, rather than planar or banded as predicted by theory.

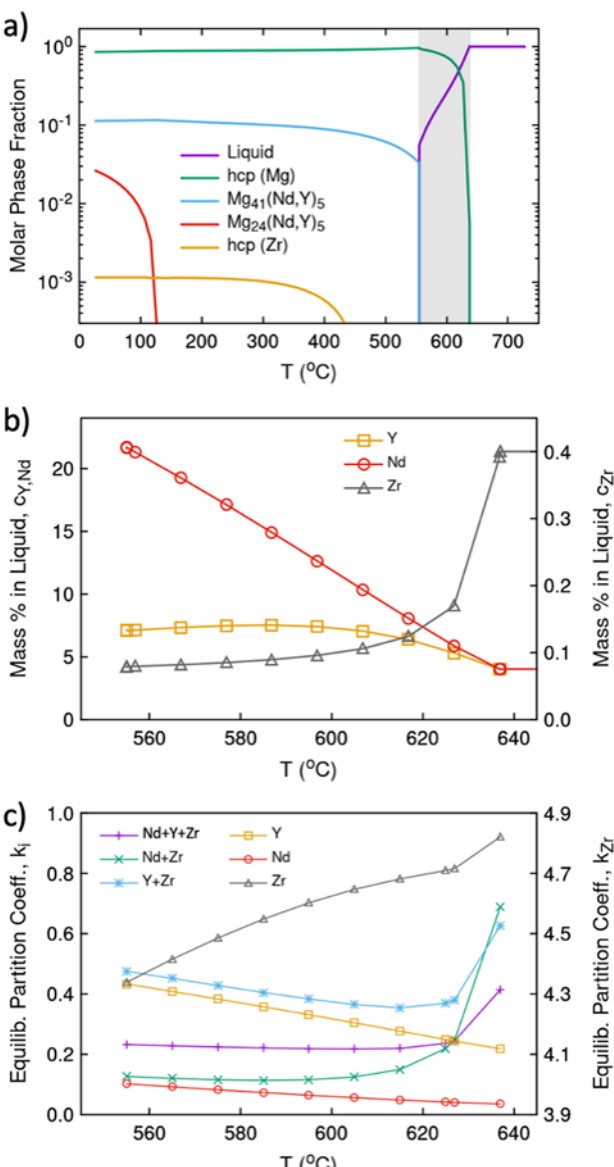

**Figure 10.** Computational thermodynamic analysis of WE43 solidification in equilibrium. (**a**) Property diagram showing molar-phase fractions, with the shaded area illustrating the solid–liquid coexistence region. (**b**) Elemental mass % on the liquid side of the solid–liquid interface. (**c**) Partition coefficient across the solid–liquid interface ($c^s/c^l$).

We rationalize these findings as follows. The fine-grained equiaxed dendritic morphologies under the 50 μm beam generally correlated with keyhole-mode melting and relatively deep, non-compact vapor-depression and melt-pool shapes (i.e., Figures 5c–f and 6c–f). The cellular and planar-banded morphologies under the 80 μm beam generally correlated with conduction-mode melting and relatively shallow, compact, melt-pool shapes (i.e., Figure 6b). This indicates that the KGT theory is applicable to conduction-mode welding, but not necessarily to keyhole-mode conditions, and that in this system, heterogeneous nucleation is significantly amplified in the keyhole mode. Such nucleation could result from stronger nonequilibrium effects (with, e.g., a more turbulent flow or dendrite-tip breaking) or from differences in the melting of the $Y_2O_3$ or MgO phase that create particles favorable to the nucleation of the primary hcp phase. The effects of highly nonequilibrium keyhole-mode conditions on solidification morphology are in general poorly understood and present a set of open challenges in AM research.

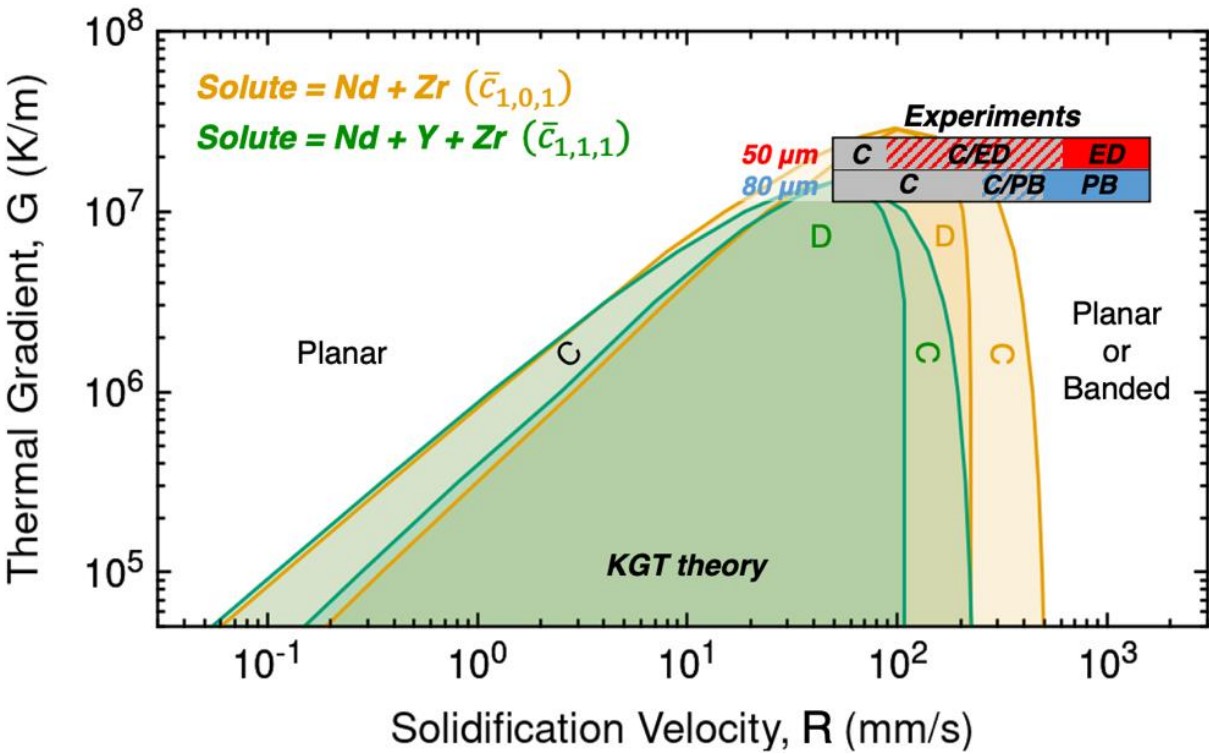

**Figure 11.** Computed WE43 solidification morphology map in *G-R* space, compared with experimental results (upper right). Results of KGT theory are presented for the two limiting cases of Y behavior (fully dispersed and nonmelted). The position of the experimental results along the *G* axis is a rough estimate and *R* is set to the scan velocity *v*. C = cellular, ED = equiaxed-dendritic, PB = planar-banded.

## 4. Conclusions

In this work, we investigated the behavior of WE43 under a wide variety of LPBF-processing conditions using in situ, high-speed X-ray imaging, and ex situ cross-sectional microscopy. In situ X-ray imaging revealed five distinct regimes of vapor-depression behavior. It revealed that for scan speeds above 500 mm/s, we did not observe any keyhole-induced porosity. Cross-sectional light microscopy indicated five distinct regimes of melt-pool geometry that roughly corresponded to the five vapor-depression regimes and allowed the identification of a laser-parameter space suitable for printing. SEM and STEM EDS elemental imaging revealed three distinct microstructural regimes that could be controlled by modifying laser-scan parameters: cellular, equiaxed dendritic, and banded microstructures. These three regimes provided an insight into the differences in the solidification rate and direction resulting from varying LPBF-processing conditions. The solidification morphologies of conduction-mode weld tracks were consistent with those predicted by analytic theory, while keyhole-mode tracks exhibited anomalous equiaxed dendritic morphologies at high velocities. The latter was attributed to enhanced heterogeneous nucleation driven by strong nonequilibrium flow effects. Secondary-phase particles rich in Y and powder-particle oxide shell remnants were identified within all tracks. There were no significant compositional differences measured within the matrix of the tracks. By mapping the response of WE43 to a variety of LPBF-processing conditions, this work serves as a guide to tailor microstructures and achieve good-quality builds.

**Supplementary Materials:** The following supporting information can be downloaded at: https://www.mdpi.com/article/10.3390/cryst12101437/s1, Video S1: In situ X-ray imaging of LPBF performed on Mg alloy WE43 using 150 W laser power, 50 μm beam diameter, and 100 mm/s scan speed, revealing the vapor depression, melt-pool boundary, formation of pores, and ejection of powder particles; Video S2: In situ X-ray imaging of LPBF performed on Mg alloy WE43 using 50 W laser power, 50 μm beam diameter, and 250 mm/s scan speed, revealing a thin, shallow vapor depression; Video S3: In situ X-ray imaging of LPBF performed on Mg alloy WE43 using 200 W laser power, 50 μm beam diameter, and 250 mm/s scan speed, revealing a deep, thin vapor depression with large, bulbous protrusions; Video S4: In situ X-ray imaging of LPBF performed on Mg alloy WE43 using 250 W laser power, 50 μm beam diameter, and 500 mm/s scan speed, revealing a thin, mid-depth vapor depression with a persistent oscillating protrusion to the rear; Video S5: In situ X-ray imaging of LPBF performed on Mg alloy WE43 using 350 W laser power, 50 μm beam diameter, and 750 mm/s scan speed, revealing a thick, mid-depth vapor depression with a persistent oscillating protrusion to the rear; Video S6: In situ X-ray imaging of LPBF performed on Mg alloy WE43 using 400 W laser power, 50 μm beam diameter, and 1250 mm/s scan speed, revealing a thick, mid-depth vapor depression with a trailing tail opening up to the surface.

**Author Contributions:** Conceptualization, J.S., N.P.C., R.E.S., J.F.L. and M.J.M.; methodology, J.S., A.A.M., N.P.C., B.V., R.E.S., J.M.B., A.P., J.N.W., K.H.S. and C.J.T.; software, J.M.B. and A.P.; validation, A.A.M., N.P.C., P.J.D., J.W., B.V., R.E.S., J.M.B. and A.P.; formal analysis, J.S., A.A.M., B.V., R.E.S., I.B. and J.M.B.; investigation, J.S., A.A.M., N.P.C., P.J.D., J.W., B.V., I.B., V.T., A.Y.F., A.M.K., J.M.B. and A.P.; resources, M.F.T., A.V.B. and J.F.L.; data curation, J.S. and J.M.B.; writing—original draft preparation, J.S., A.A.M., N.P.C., R.E.S., B.V., J.M.B., A.P. and I.B.; writing—review and editing, J.S., A.A.M., N.P.C., R.E.S., J.M.B., A.P., J.F.L. and M.J.M.; visualization, J.S., A.A.M., I.B. and J.M.B.; supervision, J.F.L., S.H.R. and M.J.M.; funding acquisition, J.F.L., S.H.R. and M.J.M. All authors have read and agreed to the published version of the manuscript.

**Funding:** The material presented here was supported by Laboratory Directed Research and Development grant 18-SI-003 from Lawrence Livermore National Laboratory (LLNL). This material is also based upon work supported by the U.S. Department of Energy's Office of Energy Efficiency and Renewable Energy (EERE), under the Advanced Manufacturing Office, CPA Agreement Numbers 32035, 32037, and 32038. LLNL is operated by Lawrence Livermore National Security, LLC, for the U.S. Department of Energy, National Nuclear Security Administration under Contract No. DE-AC52-07NA27344. Use of the Stanford Synchrotron Radiation Lightsource, SLAC National Accelerator Laboratory is supported by the U.S. Department of Energy, Office of Science, Office of Basic Energy Sciences under Contract No. DE-AC02-76SF00515. J.S. acknowledges the support received from the University of California Lab Fees Research Program, Award ID: LGF-17-476556.

**Data Availability Statement:** The data presented in this study are available upon request from the corresponding author.

**Acknowledgments:** The authors acknowledge the experimental assistance received from Doug Van Campen and Matthew Latimer at the SLAC National Accelerator Laboratory. The Scientific Center for Optical and Electron Microscopy (ScopeM), ETH Zurich, is acknowledged for providing access to the instruments.

**Conflicts of Interest:** The authors declare no conflict of interest. The funders had no role in the design of the study; in the collection, analyses, or interpretation of data; in the writing of the manuscript; or in the decision to publish the results.

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
