# Peer review of "Melt-Pool Dynamics and Microstructure of Mg Alloy WE43 under Laser Powder Bed Fusion Additive Manufacturing Conditions"

_crystals, doi:10.3390/cryst12101437_

Round 1

Reviewer 1 Report

The paper elaborates Magnesium-based alloy WE43 as a state-of-the-art bioresorbable metallic implant material. There is a need for implants with both complex geometries to match the mechanical properties of bone, as well as a refined microstructure necessary for controlled resorption. Additive manufacturing (AM) using laser powder bed fusion (LPBF) presents a viable fabrication method for implant applications, as it offers near net shape geometrical control, allows for geometry customization based on an individual patent, and the fast-cooling rates necessary to achieve a re-fined microstructure. In this study, the laser-alloy interaction is investigated over a range of LPBF relevant processing conditions to reveal melt pool dynamics, pore formation, and the microstructure of laser-melted WE43. In situ X-ray imaging reveals distinct laser-induced vapor depression morphology regimes, with minimal pore formation at laser scan speeds greater than 500 mm/s. Optical and electron microscopy of cross-sectioned laser tracks reveals three distinct microstructural regimes that can be controlled by adjusting laser scan parameters: columnar, dendritic, and banded microstructure. These regimes are consistent with those predicted by analytic solidification theory for conduction mode welding but not for keyhole mode tracks. The results provide insight into the fundamental laser-material interactions of WE43 alloy under AM processing conditions and are critical for the successful implementation of LPBF-produced WE43 parts in biomedical applications.

I recommend paper for printing and publishing 

Reviewer 2 Report

This manuscript provides extensive analysis of WE43 alloy under various AM processing conditions, which would be helpful to determine experimental parameters. Also, the description of nonequilibrium keyhole mode conditions on solidification was interesting. I would recommend publishing this manuscript if the following comments are reflected in the revised version.

1)    Not enough explanation about Figs. 4(b,c). Also, it seems that the normalized enthalpy (beta) and pore depth & size do not have any relationship between each other. What can we learn from this plot? In addition, are the parameters which determine the beta value independent or related? If related, please provide examples.

2)    In Fig. 7b, there might be a strong temperature gradient which facilitates directional growth. Despite that, is it reasonable to say that the non-equilibrium effects are dominant providing effective heterogeneous nucleants? Can you provide more evidence (or relevant citations)?

In line 623-624, isn’t the high G/R indicating planar solidification front, not equiaxed grain structures? Also, in line 625-626, the banded microstructure seen in the tracks scanned with an 80 μm beam indicates a higher G/R. Sounds a bit confusing.

3)    In Fig. 7e, was the planar banded interface structure originated from the fast growth beyond the limit of absolute stability? If so, please provide references.

Also, several minor comments:

- Line 79, “the presence of lack of fusion defects” sounds a bit confusing, what about “the presence of defects, such as lack-of-fusion pores”

- Line 82-83, While LPBF processing of WE43 has been demonstrated [citation is missing],

- Line 443, probably 5b? Cannot find a strong linear relationship in 4b.

- Fig. 8, missing inverse pole figure color key and experimental conditions.
